# Improving IoT Predictions through the Identification of Graphical Features

**DOI:** 10.3390/s19153250

**Published:** 2019-07-24

**Authors:** Syeda Akter, Lawrence Holder

**Affiliations:** School of Electrical Engineering and Computer Science, Washington State University, Pullman, WA 99163, USA

**Keywords:** sensor networks, graph representation, graphical features, feature selection, activity recognition

## Abstract

IoT sensor networks have an inherent graph structure that can be used to extract graphical features for improving performance in a variety of prediction tasks. We propose a framework that represents IoT sensor network data as a graph, extracts graphical features, and applies feature selection methods to identify the most useful features that are to be used by a classifier for prediction tasks. We show that a set of generic graph-based features can improve performance of sensor network predictions without the need for application-specific and task-specific feature engineering. We apply this approach to three different prediction tasks: activity recognition from motion sensors in a smart home, demographic prediction from GPS sensor data in a smart phone, and activity recognition from GPS sensor data in a smart phone. Our approach produced comparable results with most of the state-of-the-art methods, while maintaining the additional advantage of general applicability to IoT sensor networks without using sophisticated and application-specific feature generation techniques or background knowledge. We further investigate the impact of using edge-transition times, categorical features, different sensor window sizes, and normalization in the smart home domain. We also consider deep learning approaches, including the Graph Convolutional Network (GCN), for the elimination of feature engineering in the smart home domain, but our approach provided better performance in most cases. We conclude that the graphical feature-based framework that is based on IoT sensor categorization, nodes and edges as features, and feature selection techniques provides superior results when compared to the non-graph-based features.

## 1. Introduction

We seek to leverage the inherent graph structure in IoT sensor networks to find a generic graph representation of sensor network data. We can provide a tool to IoT application builders to boost prediction task performance if we can build a generic graph-based framework and graphical features to support different prediction tasks from different IoT sensor network data. Graph-based approaches have been successful in several IoT applications. For example, George and Shekar [1] used a graph representation for road navigation to efficiently find the shortest route that may change with time. Graph-based methods that were applied to activity recognition by Long and Holder [2] showed uncorrelated errors when compared to non-graph methods, which indicated the potential for improved performance through the use of a graph representation. Salomon and Tirnauca [3] learn users’ behavioral patterns while using weighted finite automata by building a time-aggregated graph from the activity flow of a session (morning, afternoon, evening). They represent each activity as a node and time spent for a particular activity or the time elapsed between different activities as edge labels. George et al. [4] proposed Spatio-Temporal Sensor Graphs (STSG) to capture time dependence of sensor network parameters. They generalized time-aggregated graphs and included the probability parameters to represent the stochastic nature of sensor graphs. They illustrated three case studies of anomaly detection, basic hotspot detection and growing hotspot detection while utilizing the STSG model. Based on these works, we hypothesize that representing the sensor data as a graph and providing this graphical input to graph mining algorithms to identify new features in the data can improve the performance of learning methods in recognition and prediction tasks that are related to sensor networks. We develop a Graphical Feature-based Framework (GFF) to represent and analyze IoT sensor network data. This framework collects data from sensor networks, uses graph structure to represent movement-related data, and employs selected graphical features to improve the corresponding prediction tasks. As part of this framework, the sensors are represented as nodes in a graph, and consecutive triggering of sensors are represented as edges in the graph. Subsequently, graphical features, such as existence of nodes, existence of edges, count of nodes, and count of edges, are extracted from the graph to construct a feature set. This graphical feature set is added to the typical feature set of the IoT application. This augmented feature set is supplied to a classifier to learn a model to perform the prediction task for the IoT application.

We applied the GFF to three different IoT domains with different sensors and different prediction tasks. The first domain is activity recognition from smart home motion sensors, the second domain is demographic information prediction from smart phone GPS sensors, and the third domain is activity recognition from smart phone GPS sensors. Our graphical feature-based approach improved the classification accuracy for the first two domains as compared to most of the benchmarks in each domain [5,6]. For the third domain, we use graphical features along with typical non-graphical features that are based on sensor data, such as accelerometer and gyroscope, to predict activities. Experimental results show that the inclusion of GPS data as node features significantly improves the performance over other typical non-graphical features. Analyzing the confusion matrix shows that the addition of edges may improve the performance for some activities. Using only selected features has the potential to improve the performance with the addition of edges [7]. In this article, along with graphical features used in our above-mentioned work, we propose and evaluate additional graphical features as part of our framework. Here, we assess the use of different window sizes, adding edge transition time, categorization of nodes and edges, normalization of nodes and edges, application of different classifiers, and impact on different evaluation measures for the smart home domain. We also evaluate the use of deep learning and Graph Convolutional Networks to classify activities in the smart home domain in order to compare our framework with these deep learning approaches.

In Section 2, we describe related work that was conducted in the three domains of IoT applications to which we applied our GFF. In Section 3, we describe details of each step in the graphical feature-based method that was designed for IoT applications. In Section 4, we summarize the performance of the GFF in three domains. Afterwards, we describe the implementation details of varying window sizes, normalization of non-graphical and graphical features, using additional features, such as edge transition time and categorical features, use of Linear Support Vector Machine (SVM) and Random Forest as the classifier, and the application of deep learning approaches in the smart home domain. We present the results and the impact of these experiments on the performance of the prediction task. In Section 5, we conclude with a summary of our findings and suggest the best practices for applying graphical features. In Section 6, we discuss the future directions for using graph-based representations for IoT applications. 

## 2. Related Work

In this section, we discuss previous work conducted in the three domains of activity recognition from smart home sensors, demographic prediction from mobile phone sensors, and activity recognition from smart phone sensors. Researchers have approached the problem of activity recognition from motion sensors using various methods, such as Bayes Belief Networks, Artificial Neural Networks, Sequential Minimal Optimization, LogitBoost Ensemble [8], Naive Bayes, Hidden Markov Models [9,10], and Conditional Random Fields [11]. Extensive research has been conducted regarding improving activity recognition performance while using non-graphical features such as motion sensor triggering; temporal information during the activity, such as time of day, day of week, whether the day is a weekend or not; activity length in time; previous/next activity; number of kinds of motion sensors involved; total number of times motion sensor events triggered; and, energy consumption for an activity [8,11].

Aicha et al. [12] detected the presence of multiple people or visits while using the number of sensor transitions that are not topologically connected. Two sensors are considered to be topologically connected if one person can activate those two sensors consecutively without activating another sensor. They also considered the number of sensor transitions where at least one of the sensors is a door sensor. To handle these two heterogeneous information streams with Markov Modulated Poison Process (MMPP), the authors introduced a variant of the process, called Markov Modulated Multidimensional Poison Process (M3P2), which performed better than MMPP. Their features are similar to our graphical features in regard to using sensor transitions. The difference is that we are considering any sensor transitions that occur in the data, regardless of topological connection and the types of sensors involved. Our model is applicable to multiple applications while using different kinds of sensors and prediction tasks. The model that was proposed by Aicha et al. [12] needs to be evaluated for its ability to generalize to other sensor applications and other prediction tasks. In their model, two variables representing the counts of two kinds of sensor transitions are represented as states of time. In our approach, we used the sensor transition times as additional edge features and directly feed these features to classifiers.

Mo et al. [13] used both raw data from all available sensors and applications in the mobile phone, computed tens of thousands of features through a complicated feature engineering process, designed specific features for specific tasks, and used background information, such as task-interdependency, to obtain a high performance. Nadeem et al. [14] used SVM and Random Forest with a bagging scheme over mean, variance, frequency, and many other measures to produce thousands of features that employ call log, visited GSM cells, visited Bluetooth devices, visited wireless LAN devices, accelerometer data, and other information. Brdar et al. [15] represented the data as a graph, where each node is a user, and used k-nearest-neighbor with feature selection to achieve better performance for predicting job-types. Dong et al. [16] predict the gender and age-group from call log and SMS data based on the relationship among populations of different genders and age groups. Ying et al. [17] address the imbalance class problem in the data through the use of multi-level classification and compute features that are related to users’ behavior and environment. Demographics predictions also have been done while using blogger’s writing style [18] and web-page click through data [19]. In “Limits of Predictability in Human Mobility” [20], the findings show that users’ daily mobility has a regularity and it has 93% predictability on average. Hence, it might be possible to build accurate predictive models that are dependent on human mobility. Montjoye et al. [21] emphasize the privacy concerns of an individual’s mobility due to the evidence that the uniqueness of human mobility traces is high and that mobility datasets can be re-identified while only using a few outside locations. Our GFF uses more generalized information of location categories, avoids complicated feature engineering, and uses graphical features that are only derived from GPS sensor data. We discuss more detailed comparisons to related works in the Results section.

Bouchard et al. [22] used raw latitude and longitude as the features to improve activity recognition performance and also discussed the potential of distance, position, shape, and gesture as features. Aminikhanghahi et al. [23] developed an approach, called Thyme, for adapting prompt timing that is based on the context of the user’s activity. Liao et al. [24] attempted to recognize activities from GPS traces through training a conditional random field to iteratively re-estimate significant places and activity labels. In our work, we add graphical features to the typical features that are used in this area and evaluate the potential of obtaining improved performance, along with having a general framework for IoT applications.

Recently, combining symbolic representations of domain knowledge and data-driven learning approaches has approached the activity recognition task. Yordanova et al. [25] proposed a Computational Causal Behaviour Model (CCBM) that is based on the principle of Computational State Space Models (CSSMs). They developed these models to address noisy sensor observations and the complications caused by behavior complexity and variability that are observed in real settings. They applied this approach to a sensor dataset that was collected by monitoring unscripted kitchen activities in order to detect eating behavior and provide indicators of health status. The performance was comparable to Hidden Markov Models and decision tree models. Olaru and Florea [26] used context graphs, context patterns, and continuous, persistent context matching to recognize context in ambient assisted living situations viewed as a set of associations between different concepts. Hao et al. [27] proposed a real-time inference engine while using formal concept analysis to graphically represent the concepts that can be used to improve the recognition of sequential, interleaved, and concurrent human activity patterns. Chen et al. [28] proposed an approach to address the cold-start, reusability, and incompleteness of data-driven activity modeling. They applied an ontology-based hybrid approach in three phases. In the first phase, the cold-start problem is resolved by creating seed activity models that are based on the domain knowledge that most activities of daily living (ADLs) are daily routines and normally take place at a specific time, location, and space. In the second phase, activity models are created in two levels of abstraction starting with generic coarse-grained activity models that are applicable for all users and then iteratively creating individual activity models, thus solving the reusability problem. In the third phase, the incompleteness problem is resolved by using learned activity patterns to update the activity models. Ye et al. [29] conducted similar work of applying a hybrid approach, who first use a general ontology model to represent domain knowledge across different environments and users. They then combine learning techniques with ontological semantics for unsupervised discovery of patterns for each individual’s daily activities.

In previous work, we proposed a graph-based representation of sensor data from which we extract graphical features that can improve the performance of prediction tasks for sensor networks, regardless of prediction tasks and domains [5,6,7]. We applied this concept to three different domains. Our graphical feature-based approach outperformed four previously explored graph-based approaches for activity recognition from motion sensor data in smart homes in [2]: Graph SVM, Subgraph SVM, Nearest Neighbor, and an ensemble of these three. The graphical features also outperformed three widely used baseline approaches in this domain: Naïve Bayes, Hidden Markov Model, and Conditional Random Field for the Cairo, Aruba, and Tulum smart home datasets [5]. In the second domain on demographic prediction, we used GPS location data from the Nokia Mobile Data Challenge 2012 dataset [30] to perform the classification task of demographic attributes, in particular classifying gender (two classes), age group (eight classes), and job type (eight classes). Graphical features or a combination of graphical and non-graphical features performed significantly better than only non-graphical features for all of the prediction tasks. In the third domain, activity prediction from smart phone data, the addition of GPS location category features to typical smart phone sensor-based features provided the best performance.

More recently, deep learning has been applied to the activity recognition task that is based on sensor data [31]. Like our approach, deep learning avoids the need for manual feature engineering. However, deep learning approaches can be overconfident in their classification, even for incorrect classifications, and information regarding the user’s context is not easily extracted from the learned network. To address these issues, Rueda et al. [32] proposed a hybrid activity recognition architecture, called Hybrid Computational Causal Behaviour Model (HCCBM), which combines deep learning with symbolic models, namely, CSSMs. However, deep learning has not been applied to our graph-based representation of sensor data. The Graph Convolutional Network (GCN) [33] is a deep learning approach to classification of graphs, and so it can be applied to our setting to evaluate a deep learning alternative to our GFF. The Results section presents an empirical comparison of the approaches.

## 3. Graphical Features-Based Framework

We propose a Graphical Feature-based Framework (GFF), where we represent one type of sensor data in a graph, extract graphical features, apply feature selection techniques, and then apply classification methods for a prediction task. Figure 1 shows the workflow for a particular prediction task from an IoT sensor network that is based on the GFF. We describe the details of each step of the GFF in Section 3.1, Section 3.2 and Section 3.3. The main contribution of this article is proposing and evaluating additional graphical features as part of the GFF. We describe these additional features in Section 3.4. In Section 3.5 and Section 3.6, we briefly discuss the deep learning approach and the GCN to compare with our graphical feature-based approach.

### 3.1. Sensor Categorization and Graph Representation

We consider each sensor in the IoT application, map the sensor to a sensor category if that is provided and available, and represent it as a node in the graph for representing IoT sensor network data as a graph. We construct a graph for each labeled instance of the IoT application. When two consecutive sensors are turned on during a labeled task, an undirected edge is added between the two corresponding nodes in the graph. Two sensors can be consecutively triggered multiple times, that is, an edge can be triggered multiple times. We store this count of multiple edges being triggered as edge attributes. The same sensor can be consecutively triggered in some applications, and hence we allow self-loops in our graph representation to capture this information. 

We consider motion sensors as nodes for the task of activity recognition in the smart home. Each motion sensor can be mapped to a room type such as bedroom, kitchen, dining room, and living room, which are used as the sensor’s category [5]. For the tasks of demographic prediction [6] and activity recognition from smart phone sensors [7], we consider location data from the GPS of mobile phones and categorize the locations visited. We represent each location category as a node in the graph. We find the nodes in the graph for the corresponding location categories and add an undirected edge between the two nodes whenever the participant moves from one location to another location. In this way, we create a graph for each user in the demographic prediction task and a graph for each activity in the activity recognition task. We show examples of graph representations generated from motion sensor data and from GPS data in Figure 2.

### 3.2. Graphical Feature Extraction

We construct one graph for each labeled instance in the sensor network data. We construct one instance per user where we extract the existence of nodes and edges from the user’s graph as features for the demographic prediction problem. For the activity recognition problem, we construct one graph instance per activity. We use the list of all the nodes and edges triggered in all graphs as the feature set. Initially, three different kinds of graphical features have been used: existence of nodes, existence of edges, and existence of both nodes and edges. For the existence of nodes experiment, our feature set consists of all unique sensor categories that are represented as nodes in the graphs across all instances. If a node is present in the graph for a user, we set the value to ‘ON’ for that feature in the corresponding instance; otherwise, we set the value to ‘OFF’. For the existence of edge experiment, we compute all the unique edges triggered by all instances in their sensor-trajectory graph. We use this list of all edges as the feature set for the existence of edge experiment. To construct an instance, we consider the graph for each instance and check the existence of the edges. If the edge exists, then we mark the value as ‘ON’; otherwise, ‘OFF’. In the third experiment, we combined existence of both nodes and edges that we used as features in the previous two experiments as the combined feature set. We also computed how many times each node has been turned on during an activity and used these counts as features. We store the number of times that each edge is triggered as edge attributes and use these edge counts as graphical features. We attempted to introduce additional transitional features for the demographic prediction problem. We save the types of sensors as node attributes, and we denote the time used to move from one sensor to another sensor as the edge transition time and save it as an edge attribute. We use these node and edge attributes as additional graphical features and assess their impact on prediction task performance.

### 3.3. Feature Selection and Classification

The number of features may be large when compared to the number of instances since we are using nodes and edges triggered across all graphs as our feature set. Feature selection is employed for reducing the number of features to avoid overfitting in this condition. In the literature, there are two main approaches for choosing useful and relevant attributes: filtering and wrapper [34]. The filtering approach ranks each individual feature based on different measures, such as information gain, correlation, gain ratio, and symmetrical uncertainty. In the wrapper approach, subsets of the feature set are checked with a classification algorithm, and the subset that performs the best is provided as the optimal and selected feature subset. When the size of the feature set is large with thousands of features, the wrapper approach might be computationally expensive. For a large number of features, a hybrid approach is used, where a feature set of reduced size is constructed based on the filtering approach, and then the wrapper approach is applied to the reduced feature set to find an optimal feature subset. We then apply an appropriate classification technique on this optimally selected set of features. For the classifiers, we considered the Support Vector Machine and different tree-based classifiers. We describe the details of the feature selection algorithms and the classification techniques that provided the best result for each application in the Results section.

### 3.4. Additional Features

We experimented with additional features: normalized node counts and edge counts, edge transition times. We present the results of applying these additional features to the smart home domain. We also show the result of using sensor type and room type as categorical information for smart home sensors. We performed an experiment in the smart home domain to evaluate the impact of different window sizes on the use of graphical features while classifying activities. The results from these experiments are presented in the Results section.

### 3.5. Deep Learning Approach

We explore Deep Learning (DL) approaches as alternatives to the GFF, due to DL’s ability to construct relevant features without user intervention. The advantage of using a DL network as a classifier is that it automatically identifies relevant features and avoids the need for users to construct handcrafted features. Using DL as a classifier may also avoid the need for adding graphical features if the DL classifier can internally construct these features. We provide raw sensor data and graphical features to a DL classifier to experiment with this. Afterwards, we compare the results with our GFF.

### 3.6. Graph Convolutional Network (GCN) Approach

We also explore applying a Graph Convolutional Network (GCN) to activity prediction in the smart home. Under the umbrella of GCNs, a large number of methods have been developed to re-define convolution for graph data with the first prominent research that was presented in Bruna et al. [35]. Graph based neural networks (GNNs) have been applied in the areas of computer vision, recommender systems, forecasting in traffic networks, and molecular chemistry [33]. We follow the method that was developed by Zhang et al. [36], called Deep Graph Convolutional Neural Network (DGCNN), which directly takes a graph as input to learn a classification function and thus avoids the need of transforming the graphs to vectors. This method performed better than the existing methods on many benchmark datasets. We apply DGCNN to the graphs generated from sensor data in the smart home domain. Subsequently, we compare the results with our GFF.

## 4. Results

Here, we present results on several variations of the Graphical Feature-based Framework (GFF). These variations include using different window sizes over the sensor data, normalizing feature values, adding additional features, and using different classifiers. We also show the results on using deep learning for activity prediction mainly to compare the feature generation strengths of deep learning networks for graphs to our graph-based features.

### 4.1. Window-Based Approach with Graphical Features

In previous experiments, we used an activity as an instance for constructing graphs and extracting nodes and edges. Here, we segment the duration of an activity into smaller time windows to see the effect of window size on the accuracy of activity recognition. In a real-world scenario, we may not know when an activity instance begins and ends, so we are more likely to collect sensor data over a fixed-size time window. We divide each activity into segments based on a fixed window size and then use these smaller segments as instances to construct graphs. We compute the minimum, maximum, and average window size for each activity for each dataset to decide the range of window sizes to be varied. These statistics are presented in Table 1, Table 2 and Table 3.

We vary the window sizes for each dataset from five seconds to the average window size for all activities (Aruba 1895 seconds, Cairo 1020 seconds, Tulum 4478 seconds). We apply linear support vector machine classification (LinearSVC from Scikit Learn [37]) with 10-fold cross validation and the one-vs-rest method for this multi-class classification task. Accuracy for activity recognition is shown for increasing the window sizes for dataset ‘Aruba’ in Figure 3, for dataset ‘Cairo’ in Figure 4, and for dataset ‘Tulum’ in Figure 5. In these figures, ‘full activity’ indicates the use of the entire activity for computation of graphical features instead of segmenting the activities into windows of specific time duration. ‘Full Activity’ is depicted using a horizontal line for each case of nodes, edges, and nodes+edges.

From these plots we observe that accuracy increases with increasing window size for all types of features for Tulum and Cairo. For Aruba, the highest accuracy is observed at window size 500, and accuracy decreases with larger window sizes until full activity. For Aruba and Tulum datasets, node features performed better than edge features, but the combination of nodes and edges performed the best for all of the window sizes. Edge features performed low for small window sizes near five seconds, but the performance increased faster with the increase in window size. This indicates that larger window sizes will contain more edge transitions that help to improve prediction performance. For the Cairo dataset, accuracy in general increases with window size for all types of features, but for node and edge features, accuracy sometimes decreases with increasing window size. For some window sizes, edges performed better, and for some window sizes, nodes performed better, as depicted in Figure 4. However, the combination of nodes and edges performed better than nodes and edges separately as features. Cairo accuracies using the combination of nodes and edges are increasing for all increasing window sizes, with one exception at window size of 900 s. We can conclude from the plots in Figure 3, Figure 4 and Figure 5 that either edge counts or the combination of node and edge counts always provides better performance than node and edge separately, regardless of window size. 

When we are using varying window sizes from five seconds to average activity duration, there are cases when window sizes for some instances of some activities are smaller than the current window size under evaluation. In this case, we used the whole activity as the training or test instance. In a real-world scenario, the label of the test instances and their window length will be unknown. As a result, we have to use a window size W for all test instances. There is extra irrelevant data in the test instance if the test instance we are trying to classify is an activity whose duration is D in the training set, and D < W, which may confuse the learning model and may decrease performance. In the same way, if D > W, then some relevant data is excluded from the test instance. This experiment of varying window sizes can be re-designed using a fixed window size of W for all test instances to address these possibilities. Afterwards, we can observe how the performance of basic features changes with the change of window size and how adding graphical features to basic features improves for different window sizes. This redesigned experiment will be pursued in future work.

### 4.2. Normalization

The number of nodes and edges triggered may vary with varying duration of different activities. Thus, we considered normalizing the graphical features (node counts, edge counts, and total number of nodes and edges) with respect to the total number of sensor events that occurred during each activity. Table 4 and Figure 6 show the impact of normalized graphical features on the accuracy of activity prediction. We used linear SVM from Scikit-Learn with default parameters, including one vs. rest as the method for multi-class classification. The combination of nodes and edges performed the best for all three testbeds. For Aruba, normalized features provide the best result; for Tulum and Cairo, unnormalized features provide the best result. Node normalization significantly improved the accuracy in Aruba and slightly in Tulum. Edge normalization improved accuracy in Aruba slightly and in Tulum significantly. Node and edge normalization both decreased performance in Cairo. Normalizing the combination of node-edge increased accuracy slightly in Aruba and decreased accuracy both in Tulum and Cairo. In all tables the best performers are in bold.

According to a paired *t*-test with 95% confidence, all of the higher-accuracy results for Aruba are statistically significant. For Tulum and Cairo, unnormalized nodes+edges perform significantly better than the nodes and edges. Normalized node-edges perform significantly better than the nodes and edges for both of these testbeds. However, the performance difference between unnormalized nodes+edges and normalized nodes+edges is not significant in Cairo and Tulum. 

Models that are based on counts of nodes and counts of edges may be susceptible to imbalanced datasets. These counts may vary depending on the duration of activities. We experimented using normalization to handle these cases. For an imbalanced dataset, the variance in the counts will be lower for activities with higher frequency in the data when these features are discriminating. Variance can be higher for activities with lower frequency in the data. Normalization of these two features may help in reducing the difference in variances among different activities with varying frequencies, and thus provide improved performance for imbalanced datasets.

### 4.3. Random Forest Classifier on Smart Home Domain

We also considered using a different classifier as the base classifier in the smart home domain. Table 5 presents the performance comparisons between Support Vector Machine (SVM) and Random Forest (RF) applied to three sets of unnormalized transitional features for each testbed. The best performance is from RF with node-edge in Aruba, from SVM with node-edge in Tulum, and from RF with node in Cairo.

We performed paired *t*-test for entries in Table 5 with 95% confidence. For Aruba, node-edge with Random Forest performed significantly better than the node only and edge only with Random Forest. Node-edge with Random Forest performance is significantly better than node-edge with Linear SVM. For Tulum, the performance difference between Linear SVM with node-edge and Random Forest with node-edge is not statistically significant. For Cairo, RF with nodes performs significantly better than RF with edges or node-edges, and RF with nodes performs significantly better than the Linear SVM with nodes. 

### 4.4. Edge Transition Times as Additional Features 

In this experiment, we consider the effect on performance by including the time that is taken to traverse an edge. Edge transition time is the time that is spent to arrive at the second node of an edge after triggering the first node. We compute this transition time for each edge, store it as an edge attribute in the graph, and use this information as an additional feature, along with edge count for each edge. Each edge can be triggered multiple times in an activity graph, resulting in a list of varying transition times. We use the average of these transition times as the feature value for each edge transition time. Table 6 shows the accuracy after adding transition time to the feature list for the three testbeds: Aruba, Tulum, and Cairo. Node-edge features performed the best for Aruba and Tulum. For Cairo, node is the best performer, indicating that adding edge transition times reduced the performance of graphical features in the case of Cairo. Performing a paired *t*-test on the results of this table show that edge and node-edge performance is significantly better than node performance for Aruba. For Tulum, node-edge performance is significantly better than both node and edge. There are no significant differences in the performance among the graphical features in Cairo. These results after adding edge transition time did not improve upon our best result so far.

Table 7 shows the number of instances, nodes, edges, and combination of nodes and edges. The number of edges doubled due to adding transition time for each edge. Accordingly, we next considered the use of feature selection to see whether the use of only selected features improves performance. We used the SelectKBest feature selection method to select the k most important features based on the mutual information criteria [38,39,40]. We varied the value of *k* from 10 to the maximum number of features for each testbed. We provide this range of possible values to GridSearchCV in Scikit-Learn that conducts an exhaustive search over the specified parameter values of an estimator. The parameters of the estimator are optimized by cross-validated grid-search over a parameter grid [37]. The parameters selected are those that maximize the score of the left-out data while using accuracy as the score. We separate each testbed data into 67% training and 33% testing. Subsequently, we provide the training set to GridSearchCV to select the best value for *k* and to find a fit with the Linear SVM and Random Forest classifiers. We use the optimized *k* value and classifier to classify the testing data. Table 8 presents accuracy for this prediction task while using Linear SVM as the classifier.

We observe that Node-Edge provides the best performance for Aruba and Tulum with linear SVM and with transition times as additional features. For Cairo, Node is the best performing features in this case where edge transition times are used. We observe that a mix of node counts, edge counts, and average edge transition times are selected as the best features for the three test beds.

With Random Forest as the classifier, Edge performed best for Aruba, and Node performed best for Tulum and Cairo. Table 9 presents the results for accuracy.

Table 5 shows the accuracies before adding edge transition time. Statistical significance testing could not be easily done, given the difference in experimental settings, but comparing the above results with this previous result indicates that the best result is without using edge transition time for Tulum and Cairo. However, for Aruba, adding edge transition time and using Random Forest as the classifier provides slightly better accuracy.

### 4.5. Categorical Features

In this section, we add other types of sensors as nodes along with motion sensors. We add transitions between these different types of sensors as edges. We assign a room type to each sensor, add room types as node attributes, and use room types and transition between room types as the features. Among the three testbeds, only Aruba has door sensors. After adding door sensors, accuracy using nodes significantly improved, accuracy using edge and node-edge improved slightly. Table 10 presents this result.

All of the sensors deployed in a smart home can be grouped according to their locations in the residence. Motion and door sensors located in bedrooms, bathrooms, kitchen, dining room, hallways, and living room can be mapped to the corresponding room type. We store room type as a node attribute while constructing an activity graph. Instead of using individual sensors as nodes, we use their room types as nodes and transitions between room types as edges. We manually create the mapping between sensors and room types from the apartment layout and provide the map to the feature extraction algorithm.

Table 11 shows the accuracy using room types to create an activity graph and nodes and edges as features. We observe that the combination of nodes and edges provided the best accuracy for all test beds.

Next, we add nodes and edges that are related to room types as additional features to our previous motion and door sensor-based graphical features. Table 12 and Table 13 present accuracies while using these additional features with Linear SVM and Random Forest as classifiers, respectively. Edge or the combination of nodes and edges performed the best for all three testbeds and for both classifiers. Performing a paired *t*-test with 95% confidence on the data for Table 12 shows that, for Aruba, both edge and node-edges perform significantly better than only nodes. However, the performance difference between edges and node-edges is not statistically significant. For Tulum, the node-edges perform significantly better than only nodes and only edges. For Cairo, all results are statistically significant. Conducting a paired *t*-test on the data for Table 13 demonstrates that node-edge performs significantly better than only nodes or only edges for Aruba. For Cairo, the node-edge performs significantly better than edges, but the performance difference between nodes and node-edge is not significant. For Tulum, edges perform significantly better than node-edges, but the performance difference between nodes and edges are not significant. When comparing with the previous best result shown in Table 5, for Aruba, performance slightly improved after adding room type features; for Tulum, accuracy decreased after adding room type; and for Cairo, adding room type did not affect accuracy.

Raw sensor information, such as GPS coordinates, can be too specific for learning algorithms in sensor applications. Mapping raw sensor information to categories (e.g., mapping GPS to location categories, such as coffee shop, office, home, etc.) improved the accuracy for prediction tasks compared to raw sensor information. Mapping motion sensor data in a smart home to room type information, such as bedroom, living room, kitchen, etc., and then computing graphical features on this categorized sensor information, further improves the performance of prediction tasks for single resident apartments. IoT practitioners who will use the GFF can supply mapping information from sensor id to sensor type.

We conclude that using the combination of nodes and edges as features improves the activity recognition performance in most cases when compared to nodes or edges only. Adding sensor types that are related to movement such as door sensors along with motion sensors improved the recognition accuracy. For a single resident apartment, such as Aruba, feature extraction using an activity window size of less than the average window size showed the best performance instead of using full activity time. Graphical feature extraction using full-activity as the window size provided the best performance for apartments with multiple residents, like Cairo and Tulum. In the case of single resident apartments, like Aruba, adding edge transition time combined with a feature selection method improved the accuracy of activity recognition. Adding edge transition time did not improve performance in the case of multi-resident apartments, such as Cairo and Tulum.

### 4.6. Deep Learning Approach

In these experiments, we apply a deep learning approach to activity recognition in the smart home domain. We used Keras [41], which is a high-level neural network API that is built on top of TensorFlow [42], which is an open source platform for machine learning. We used Google Collaboratory cloud service to create and train the neural net with GPU support. We created a basic sequential neural network of three dense layers. ‘Dense’ indicates fully connected layers where each node in the next layer is connected to all nodes in the previous layer. We conduct two experiments with existence of graphical features and count of graphical features as inputs to the neural network. The input layer contains 256 nodes and the hidden layer contains 128 nodes. The output layer contains one node for each activity. The hidden layer uses ReLU as the activation function and the output layer uses softmax as the activation function. We used ‘adam’ as the optimizer, ‘sparse categorical crossentropy’ as the loss function, and ‘accuracy’ as the metric. We conducted training for 30 epochs while using the train-test separation as the evaluation method with 70% data in the training set. Table 14 shows average accuracy on the test set for the three test-beds when we provide existence of nodes, edges, and combination of nodes and edges as inputs to the three-layer neural network. We present the accuracies in Table 15 resulting from the same neural network when we provide counts of nodes, edges, and combination of nodes and edges as inputs. 

We observe from the above results that edges or node-edge combination work better than only nodes for the DL approach for most of the cases, except for the single-resident apartment Aruba with count features. For Aruba, nodes performed the best with the DL approach. Count features performed better when compared to existence features for the DL approach, similar to the non-DL approach with graphical features. In Table 16, we present the best results from graphical features and the combination of features and algorithm that provided the best result in our GFF approach. Statistical significance testing could not be easily done, given the different in experimental settings, but comparing these performances with the DL accuracies in Table 15, we observe that graphical features perform similar to the DL network for single resident apartment Aruba. The graphical features performed better than the DL network for the multi-resident apartments Cairo and Tulum.

### 4.7. Graph Convolutional Network (GCN) Approach

We used the Graph Convolutional Neural Network (GCNN) that was developed by Zhang et al. in order to test the ability of deep learning to classify our graph representation of activities directly [36]. We generated the graphs for each test bed and provided those to the GCNN tool that trains the neural net for 500 epochs. We use the train and test separation method as the evaluation method with 70% data in the training set. We report the test set accuracy on the 500^th^ epoch. The GCNN tool also has an option to take continuous node attributes as inputs along with node and edge information. Here, we conduct two different experiments: (1) provide input graphs with nodes and edges information and (2) provide input graphs with nodes, edges, and node attributes indicating the count of how many times each node is triggered. We present the results of these two experiments in Table 17 and Table 18. 

We observe from Table 17 and Table 18 that GCNN with node count as attributes performs better than GCNN with existence of nodes and edges for multi-resident apartments Cairo and Tulum, but GCNN with node count performs worse for single resident apartment Aruba. When comparing with the best results from the GFF, as shown in Table 16, we observe that the GFF performs better than GCNN.

## 5. Conclusions

We studied and assessed the use of graph representations and graphical features in IoT sensor networks to improve the performance on prediction tasks. We proposed a graphical feature-based framework (GFF) to apply to sensor network data. The purpose of the construction and use of this kind of framework is multi-fold. First, the framework leverages inherent graph structure of sensor networks to represent sensor network data. Second, the framework offers a generic approach of applying graphical features to improve the accuracy on prediction tasks across different sensor networks. Third, the framework improves the predictions without using application-specific and prediction-task specific feature crafting.

We evaluated this hypothesis on three application areas, activity recognition from motion sensors in smart homes, demographic prediction from smart phone GPS, and activity recognition from smart phone GPS. The GFF approach improved the accuracy of activity recognition from motion sensors in most cases when compared with other widely used baseline methods for motion sensor data, such as Hidden Markov Model, Naïve Bayes, and Conditional Random Field. For demographic attribute prediction, the GFF outperformed most of the state-of-the-art methods while using no background knowledge and while using no application-specific and task-specific feature construction. The results for activity recognition from GPS data demonstrate that adding edges can increase the accuracy for most activities.

We present the results of using graphical features from different window sizes, while using normalized and categorized graphical features, applying edge transition times as graphical features and comparing Random Forest with Linear SVM for motion sensor data in a smart home. We analyzed the impact of graphical features on activity recognition accuracy from motion sensor data for varying window sizes. For each window size, either the edges or the combination of nodes and edges as features performed better than using only nodes. For the single-resident apartment, slight improvements in accuracy were observed by each of normalizing graphical features, adding graphical features that are based on room types, adding edge transition times as additional features, and using important features with feature selection methods. For multi-resident apartments, these additional features, along with feature selection, did not show improved accuracy. For activity recognition using smart home motion sensor data, SVM worked better for some testbeds and Random Forest worked better for other testbeds. We also presented a comparison between the GFF, a deep learning approach, and the Graph Convolutional Network, with the initial results indicating that the GFF outperforms both of these methods.

## 6. Future Directions

This study suggests several directions for future research. One research direction is to extract multi-edge paths and small subgraphs along with nodes and edges as features and evaluate the effect of these graphical features on prediction tasks for IoT sensor network applications. We extracted features representing two-edge transitions for one testbed in activity recognition from motion sensors in the smart home, but the initial results showed decreased performance. Moreover, generating longer paths and larger sub-graphs can be computationally expensive and time-consuming combined with higher numbers of features. However, a more systematic exploration of multi-edge transitions with feature selection methods, and evaluating these features on different testbeds and applications, is needed to assess whether multi-edge-based features is a promising direction. We would also like to compare our approach with symbolic representation-based approaches discussed in the related work section and to assess the possibility to combine the methods for improving the performance of sensor network prediction tasks. Symbolic approaches and hidden Markov models exploit the structure of high-level activities, and our approach exploits the sensor’s topology. We can analyze whether there are similar structures in both cases and if the combination of high-level activity structure with low-level sensor structure improves the performance of the model.

We plan to try classifiers other than SVM and Random Forest for improving prediction task performance and evaluate the GFF. We would like to investigate the reasoning behind why different classifiers are working better for different datasets and applications. Based on that, we would like to provide the best practices for classifier selection in the GFF.

For continuous features, like node and edge counts, we can transform them into nominal features. Transforming a continuous variable to a binary variable is called dichotomization [43]. In our work, the existence of nodes and edges can be considered a dichotomization of the counts of nodes and edges. Using dichotomized features may result in information loss and decreased performance [44]. The choice of cut point to dichotomize the feature is another challenging issue as the amount of information lost depends on the prior choice of cut points, with the optimal cut point depending upon the unknown parameters [45]. This aligns with our result in the smart home environment, where we obtained better performance with count features instead of existence features. However, many of the feature selection algorithms provide better results with the discretized features. Discretization is used as a pre-processing step for correlation-based feature selection approaches [46]. This aligns with our result in demographic attribute prediction from GPS data, where selecting fewer and important features from more than 7000 features is required to improve the performance. In this case, discretizing the features before providing them to the feature selection algorithm was helpful. In the future, we can experiment with discretizing counts of nodes and edges into several intervals with discretization methods that were described in [46], so that information loss is minimal and at the same time feature selection methods can effectively use the discretized features. 

With time series data, forecasting practitioners usually want to predict future events that are based on historical data. As a result, cross-validation with random subsets of the data is not a good choice, because it will be easier to learn looking at the information both before and after the dates that we are trying to predict, which is different than the real-life scenario, where we need to predict the events that will happen in the future based on events that happened in the past [47]. Additionally, research shows that performance on testing data closer-in-time to the training data will likely be better than performance on testing data farther-in-time from the training data [48]. One way to address this issue is to wait and see in real time. For example, Makridakis et al. [49] made forecasts in September 1987 and evaluated them at the conclusion of 1988. Instead of waiting a long time, we can use earlier data to fit and estimate a model and reserve later data to evaluate the forecasting accuracy. This becomes structurally similar to the real-world environment, where we are in the present and predict the future. In our future work, we will evaluate based on test data that appear later in the dataset instead of a randomly-selected subset of the data for time series forecasting from IoT sensor network data.

Generally, human activities follow a pattern of nodes visited and edge transitions. Over time, activity patterns may change, and new transitions may appear in the test data that were not seen in the training data. We can evaluate the effect of these new transitions on the activity recognition accuracy by observing the performance drift of the models on new test data. Models exhibiting a significant decrease in performance over time will need to be retrained.

While deep learning approaches to activity recognition from sensor data have shown promise [31], they performed worse in our setting than our non-deep learning approach. Deep learning approaches can be improved by tuning hyper parameters, such as number of layers, learning rate, iterations, number of units in hidden layers, choice of activation function, momentum term, mini batch size, regularization parameters, and many others. However, significant network tuning would eventually equate to the effort of feature-crafting in the traditional machine learning approach that our framework was designed to avoid.

The usefulness of the GFF can be assessed in other IoT scenarios that are widely used for tracking and monitoring. We built three different applications for data processing and modeling for three different domains. In the future, we can build one single software application and define a general interface for all IoT data, so that sensor information, sensor id to sensor category mappings, and other inputs can be provided as part of this input interface. As a result, any IoT sensor network data can be fit into this input format and easily tested to evaluate the performance improvement provided by the use of graphical features.

## Figures and Tables

**Figure 1 sensors-19-03250-f001:**
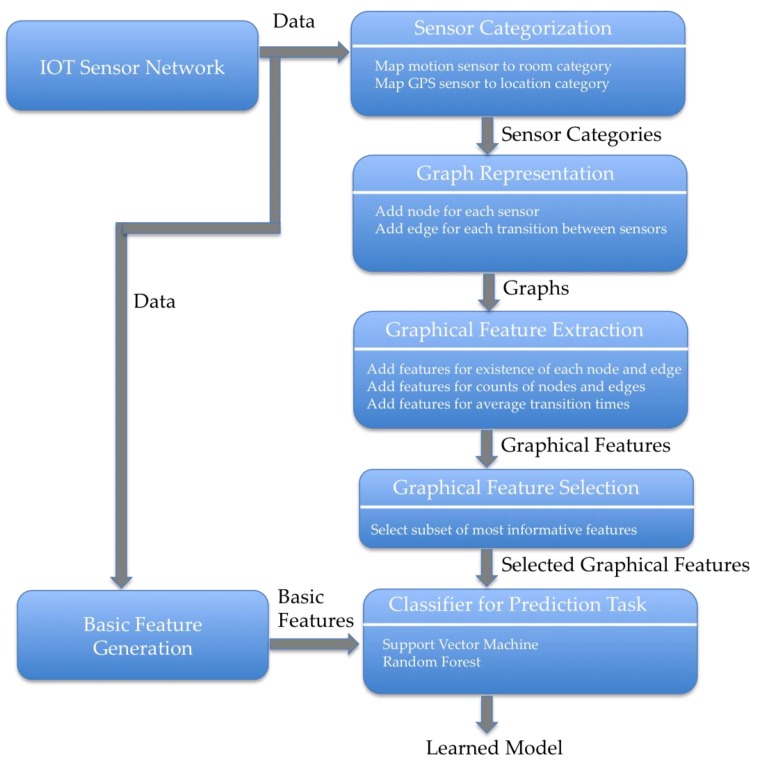
The Graphical Feature-based Framework (GFF) for IoT sensor network prediction tasks.

**Figure 2 sensors-19-03250-f002:**
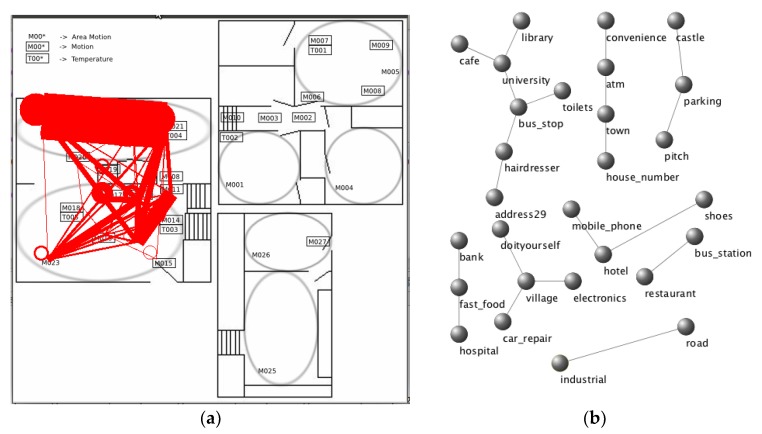
Example graph representations: (**a**) graph representation for activity ‘Breakfast’ in testbed ‘Cairo’; and, (**b**) selected edges that are triggered for a participant of age 22–27 in the Nokia smartphone dataset.

**Figure 3 sensors-19-03250-f003:**
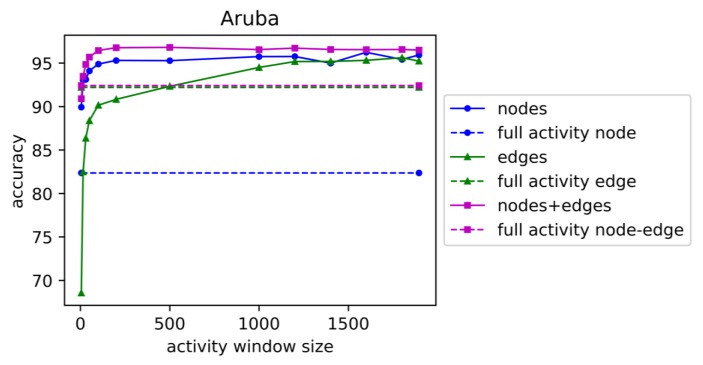
Accuracy of activity recognition for varying window sizes for testbed ‘Aruba’. ‘nodes’, ‘edges’ and ‘nodes+edges’ indicate the types of features extracted from the graph: node-based features only, edge-based features only, and both node and edge-based features, respectively. ‘full activity’ means the window size is set to the whole duration for an activity for building the graph.

**Figure 4 sensors-19-03250-f004:**
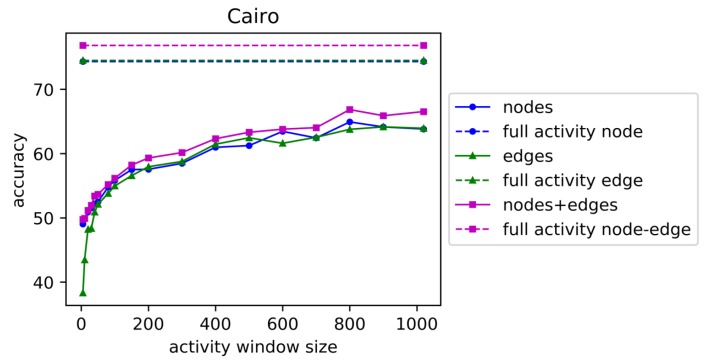
Accuracy of activity recognition for varying window sizes for testbed ‘Cairo’. See Figure 3 caption for explanation of different plots.

**Figure 5 sensors-19-03250-f005:**
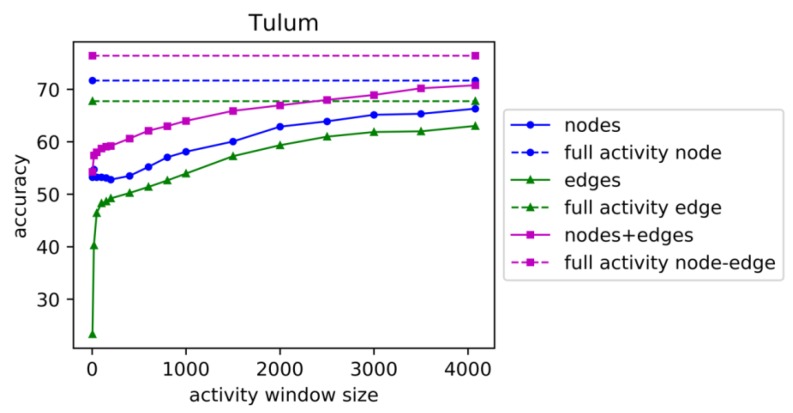
Accuracy of activity recognition for varying window sizes for testbed ‘Tulum’. See Figure 3 caption for explanation of different plots.

**Figure 6 sensors-19-03250-f006:**
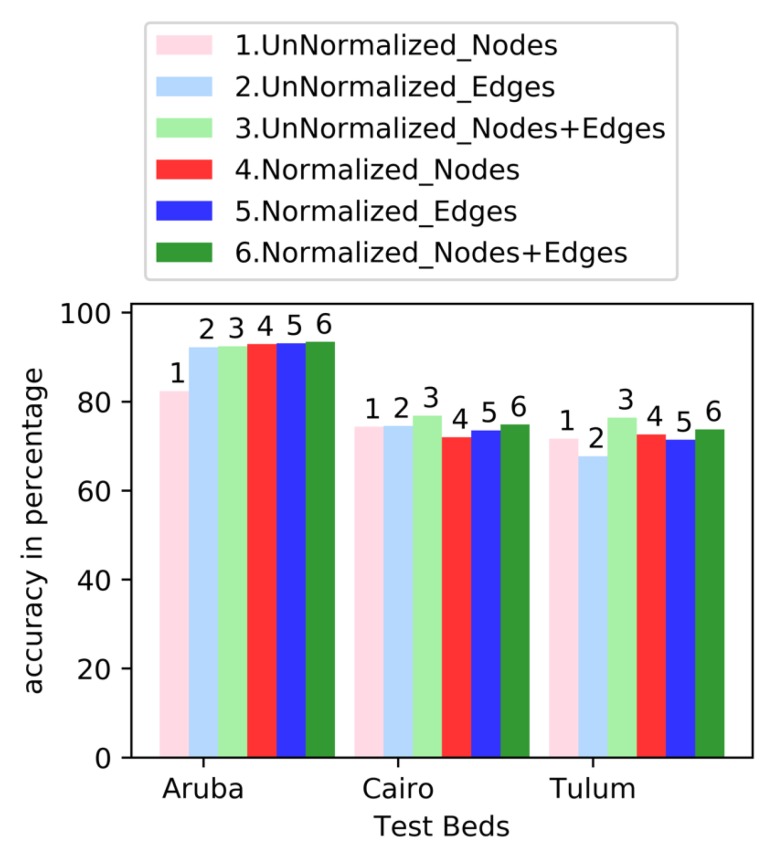
Accuracy with Normalized vs. Unnormalized Features.

**Table 1 sensors-19-03250-t001:** Aruba Dataset: minimum, maximum, and average window length of each activity.

Activity	Minimum	Maximum	Average
Respirate	97	1493	512
Enter home	1	54	5
Work	50	7016	1024
Eating	14	4167	608
Housekeeping	92	5399	1216
Relax	56	15,303	2009
Wash dishes	34	2200	428
Sleeping	30	38,783	14,417
Leave home	1	109	3
Bed to toilet	22	520	163
Meal preparation	7	19,573	469

**Table 2 sensors-19-03250-t002:** Cairo Dataset: minimum, maximum, and average window length of each activity.

Activity	Minimum	Maximum	Average
Night wandering	79	1701	298
Medicine	27	398	82
Laundry	225	807	468
Work	421	5613	2029
Bed	3	2517	659
Lunch	611	5152	1920
Dinner	1000	6236	2757
Bed to toilet	91	703	238
Leave home	3	201	20
Breakfast	551	2887	1735

**Table 3 sensors-19-03250-t003:** Tulum Dataset: minimum, maximum, and average window length of each activity.

Activity	Minimum	Maximum	Average
Bathing	1	53,360	270
Wash dishes	1	828	119
Personal hygiene	1	2343	138
Leave home	1	1348	21
Enter home	1	54,099	477
R1 sleeping in bed	202	123,772	32,463
R2 sleeping in bed	1000	123,776	34,490
Work bedroom 1	1	14,011	548
Work table	1	5100	147
Watch TV	1	10,100	362
Meal preparation	1	64,968	169
Yoga	26	3302	827
Eating	2	11,156	404
Bed toilet transition	1	800	79
Work bedroom 2	1	27,328	550
Work living room	1	12,728	584

**Table 4 sensors-19-03250-t004:** Accuracy with Normalized vs. Unnormalized Features.

Test Bed	Node Accuracy		Edge Accuracy		Node Edge Accuracy	
	Un-Norm	Norm	Un-Norm	Norm	Un-Norm	Norm
Aruba	81.03	92.90	92.13	93.11	92.03	**93.41**
Tulum	71.51	72.62	67.95	71.40	**76.34**	73.74
Cairo	73.17	72.00	74.17	73.50	**76.50**	74.83

**Table 5 sensors-19-03250-t005:** Accuracy (Unnormalized features) Support Vector Machine (SVM) vs. Random Forest (RF).

Test Bed	Node Accuracy		Edge Accuracy		Node Edge Accuracy	
	SVM	RF	SVM	RF	SVM	RF
Aruba	81.03	92.91	92.13	93.06	92.03	**93.07**
Tulum	71.51	72.45	67.97	70.53	**76.34**	72.46
Cairo	73.17	**79.17**	74.17	75.33	76.5	77.67

**Table 6 sensors-19-03250-t006:** Accuracy with Edge Transition Time as Additional Feature.

Test Bed	Node	Edge	Node Edge
Aruba	89.04	91.76	**91.80**
Tulum	72.01	67.99	**73.38**
Cairo	**73.67**	66.17	65.67

**Table 7 sensors-19-03250-t007:** Dimension of Feature Matrices with Edge Transition Time as Additional Features.

Test Bed	Instances	Nodes	Edges	Nodes + Edges
Aruba	6477	34	976	1010
Tulum	12635	31	972	1003
Cairo	600	27	722	749

**Table 8 sensors-19-03250-t008:** Accuracy with Selected Edge Transition Time and Linear SVM.

Test Bed	Node	Edge	Node Edge
Aruba	89.71	91.16	**91.39**
Tulum	72.01	68.82	**72.69**
Cairo	**74.75**	69.19	61.62

**Table 9 sensors-19-03250-t009:** Accuracy with Selected Edge Transition Time and Random Forest.

Test Bed	Node	Edge	Node Edge
Aruba	92.70	**93.59**	93.26
Tulum	**72.13**	69.21	71.56
Cairo	**78.28**	74.24	73.23

**Table 10 sensors-19-03250-t010:** Accuracy Before and After Adding Door Sensors with Motion Sensors for “Aruba”.

Test Bed	Node	Edge	Node Edge
Motion Sensors	81.03	92.13	92.03
Motion and Door Sensors (Linear SVM)	89.49	92.62	92.87
Motion and Door Sensors (Random Forest)	**93.07**	**93.27**	**93.42**

**Table 11 sensors-19-03250-t011:** Accuracy after Mapping Sensors to Room Type.

Test Bed	Node	Edge	Node Edge
Aruba	92.73	92.65	**92.81**
Tulum	60.42	60.81	**61.98**
Cairo	73.00	75.17	**76.50**

**Table 12 sensors-19-03250-t012:** Accuracy after Adding Mapped Room Type to Motion Sensors (Linear SVM).

Test Bed	Node	Edge	Node Edge
Aruba	89.19	**92.76**	92.47
Tulum	71.95	68.25	**75.56**
Cairo	71.50	74.33	**74.67**

**Table 13 sensors-19-03250-t013:** Accuracy after Adding Mapped Room Type to Motion Sensors (RF).

Test Bed	Node	Edge	Node Edge
Aruba	93.05	93.22	**93.53**
Tulum	72.49	70.53	**72.60**
Cairo	78.00	**79.17**	77.50

**Table 14 sensors-19-03250-t014:** Accuracy from Deep Learning (DL) approach with input of existence of graphical features.

Testbeds	Nodes	Edges	Nodes + Edges
Aruba	91.87	**92.74**	92.49
Cairo	62.22	67.22	**67.22**
Tulum	69.18	69.82	**71.35**

**Table 15 sensors-19-03250-t015:** Accuracy from DL approach with input of count of graphical features.

Testbeds	Nodes	Edges	Nodes + Edges
Aruba	**93.52**	93.36	92.49
Cairo	65.56	**73.33**	72.78
Tulum	73.8	**74.09**	72.66

**Table 16 sensors-19-03250-t016:** Best results from GFF.

Testbeds	Accuracy	Features + Algorithms for the Best Result
Aruba	93.53	Nodes + edges + doors-room types as additional features + Random Forest
Tulum	76.34	Nodes + edges + Linear SVM
Cairo	79.17	Nodes + Random Forest

**Table 17 sensors-19-03250-t017:** Graph inputs with only existence of node and edges.

Test Beds	Accuracy
Aruba	70.56
Cairo	33.33
Tulum	21.11

**Table 18 sensors-19-03250-t018:** Graph inputs with nodes, edges and count of nodes triggered as node attributes.

Test Beds	Accuracy
Aruba	35.00
Cairo	36.66
Tulum	22.32

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
