# Peer review of "Improving IoT Predictions through the Identification of Graphical Features"

_sensors, 2019, doi:10.3390/s19153250_

Round 1
Reviewer 1 Report
The paper is interesting and the results is good. But there exist some flaws that make the paper not so good.
It's better to list the contributions of the paper in the introduction section.
It's better to provide the framework with more details, as the Fig1 is too simple.
In the section of results, there exists too many Tables. It's better to reduce the number of tables.
In the section of resutls, the author list too many comparision results. It's better to list the important results that could show the superiority of the proposed method.
Reviewer 2 Report
The authors present a study which addresses the use of graph representation and graphical features in IoT sensor networks to improve the performance on recognition and prediction tasks. They have achieved significant results and their experiments are well designed.
However, I still have some comments:
In Section 1, the authors did not provide sufficient background about their study. This makes it difficult for readers to define the value and significance of their research. At Line 31, a paragraph should be inserted to explain the problematic. In addition, there are not enough references to support their hypothesis and opinions (e.g. Lines 31-33, 33-36).
In Section 4, the authors detailed their excellent performance in various experiments, however, few explanations about these results have been given. Thus, It is difficult for readers to understand the reasons behind these results. As a consequence, the rules or authors’ experience cannot be reused very well.
Reviewer 3 Report
The manuscript proposes a graph-based approach for feature extraction for sensor-based classification problems (such as activity recognition). The approach is interesting as it explores the sensors' topology in order to identify relations that are later used as features for improving the classification performance. The authors have performed thorough evaluation and the results are promising. I still have some comments that I believe have to be addressed before the manuscript is ready for publication:
- In the Introduction, the authors should state more clearly what exactly they mean by graphical feature-based approach.
- In the related work, symbolic or hybrid approaches for AR should be mentioned, especially because they also are exploring a semantic (often graph-based) structure, just on a higher abstraction level (structure in the high level activities vs. structure in the sensors' topology). For example, see [1,2,3]. Also there are hybrid approaches that combine symbolic graph-based approaches with deep learning, which deserve mentioning (e.g. see [4]).
- In the Results section, Section 4.1. "Previous results", are these your previously published results, or are these results that show the comparison with existing approaches? Please, make that clear. If these are previously published results, you should move this to the related work section and make clear what is the contribution of the paper and how it differs from your previous work. In Section 4, you can then just make a table that shows your results compared to other existing approaches applied on the same datasets.
- In Tables 4-6, please explain in the caption what is node accuracy, edge accuracy, node edge accuracy, and full activity.
- In Figures 3-5, in each figure there are several lines for "full activity". When printed in grayscale, it is not clear what is the difference between these lines. It's not even in the legend. I think you should choose a colour scheme that is distinguishable in black and white, or even better use different types of dashed lines.
- Figure 6 also has problem with the colours, in grayscale they are impossible to distinguish.
- Generally, when saying that something is significantly better / different from something else, do you have empirical evidence for that (in the form of statistical tests)? Also I would have expected you to perform statistical tests in all experiments as the differences in performance are relatively small and it could be due to chance.
- It would have been interesting to compare your approach with symbolic approaches or even with an HMM as your approach exploits the structure of the sensors' topology while they exploit the structure of the high level activities. An interesting analysis would have been to see if there are similar structures in both cases and if the combination of high level activity structure with low level sensor structure improves the performance of the model.
- The proportion of the approach description compared to the rest of the paper is very small. You should add more details to the approach description as that is your selling proposition.
[1] K. Yordanova et al. Analysing Cooking Behaviour in Home Settings: Towards Health Monitoring. In Sensors, 2019
[2] L. Chen et al. An ontology-based hybrid approach to activity modeling for smart homes. IEEE Transactions on Human-Machine Systems 44, 1, 92–105. 2014
[3] J. Ye et al. USMART: An unsupervised semantic mining activity recognition technique. ACM Transactions on Interactive Intelligent Systems 4, 4, Article No. 16. 2014
[4] F. M. Rueda et al. Combining Symbolic Reasoning and Deep Learning for Human Activity Recognition. Workshops Proceedings of the IEEE International Conference on Pervasive Computing and Communications (PerCom Workshops). Kyoto, Japan. 2019
Round 2
Reviewer 1 Report
The author answered all the questions that I have listed.
And the paper improved a lot.
Author Response
Thank you for your help improving the paper.
Reviewer 2 Report
Please add some related work to help readers better understand the state of the art about the graph representation of sensor networks for activity recognition.
In Section 1, for example, after Line 40:
[1] Salomón, Sergio, and Cristina Tîrnăucă. Human Activity Recognition through Weighted Finite Automata. Proceedings of UCAmI. Vol. 2. No. 19. 2018.
[2] George, Betsy, James M. Kang, and Shashi Shekhar. Spatio-temporal sensor graphs (stsg): A data model for the discovery of spatio-temporal patterns. Intelligent Data Analysis 13.3 (2009): 457-475.
In Section 2, for example, after Line 153:
[3] Olaru, Andrei, and Adina Florea. Context graphs as an efficient and user-friendly method of describing and recognizing a situation in AAL. Sensors 14.6 (2014): 11110-11134.
[4] Hao, Jianguo, Abdenour Bouzouane, and Sébastien Gaboury. Complex behavioral pattern mining in non-intrusive sensor-based smart homes using an intelligent activity inference engine. Journal of Reliable Intelligent Environments 3.2 (2017): 99-116.
Besides, I still think that the discussion of the proposed framework in Section 4 is not sufficient. Please discuss the following questions and explain them briefly in Section 4 or 5
1. The training data can not cover all possible situations. Therefore, the question is, can the proposed framework recognize activities with unseen data (i.e. many consecutive transitions that have not appeared in the training dataset) during the test phase, why?
2. The count of nodes and the count of edges are two important graphical features, however, they are susceptible to unbalanced data during the training phase. How does the proposed framework handle with unbalanced test data?
Author Response
Thank you for the additional suggestions for improving the paper.
We added discussion of papers [1] and [2] in the Introduction (lines 35-64, changes tracked).
We added discussion of papers [3] and [4] in the Related Work section (lines 166-171, changes tracked).
For issue #1, we added a discussion in section 6 on Future Directions (lines 680-684, changes tracked).
For issue #2, we added a discussion of this issue at the end of section 4.2 on Normalization (lines 408-414, changes tracked).
Reviewer 3 Report
I believe that the authors have adequately addressed my comments.
Author Response

(The authors gave the same response as above.)
